# Elements Characterising Multicomponent Interventions Used to Improve Disease Management Models and Clinical Pathways in Acute and Chronic Heart Failure: A Scoping Review

**DOI:** 10.3390/healthcare11091227

**Published:** 2023-04-25

**Authors:** Cristina Pedroni, Olivera Djuric, Maria Chiara Bassi, Lorenzo Mione, Dalia Caleffi, Giacomo Testa, Cesarina Prandi, Alessandro Navazio, Paolo Giorgi Rossi

**Affiliations:** 1Direzione delle Professioni Sanitarie, Azienda Unità Sanitaria Locale-IRCCS di Reggio Emilia, 42122 Reggio Emilia, Italy; cristina.pedroni@ausl.re.it; 2Laurea Magistrale in Scienze Infermieristiche e Ostetriche, University of Modena and Reggio Emilia, 42122 Reggio Emilia, Italy; mione.lorenzo@gmail.com; 3Epidemiology Unit, Azienda Unità Sanitaria Locale–IRCCS di Reggio Emilia, 42122 Reggio Emilia, Italy; paolo.giorgirossi@ausl.re.it; 4Centre for Environmental, Nutritional and Genetic Epidemiology (CREAGEN), Section of Public Health, Department of Biomedical, Metabolic and Neural Sciences, University of Modena and Reggio Emilia, 41125 Modena, Italy; 5Medical Library, Azienda Unità Sanitaria Locale-IRCCS di Reggio Emilia, 42123 Reggio Emilia, Italy; mariachiara.bassi@ausl.re.it; 6Cardiology Division, Azienda Ospedaliera Universitaria di Modena, 41124 Modena, Italy; caleffi.dalia@aou.mo.it; 7UO Medicina, Ospedale Giuseppe Dossetti, Azienda Unità Sanitaria Locale di Bologna, 40053 Bologna, Italy; giacomo.testa@ausl.bologna.it; 8Department of Business Economics, Health & Social Care, University of Applied Sciences & Arts of Southern Switzerland, CH-6928 Manno, Switzerland; cesarina.prandi@supsi.ch; 9Cardiology Division, Arcispedale Santa Maria Nuova, Azienda Unità Sanitaria Locale-IRCCS di Reggio Emilia, 42123 Reggio Emilia, Italy; alessandro.navazio@ausl.re.it

**Keywords:** heart failure, disease management, clinical pathways, chronic care model, multidisciplinary

## Abstract

This study aimed to summarise different interventions used to improve clinical models and pathways in the management of chronic and acute heart failure (HF). A scoping review was conducted according to the Preferred Reporting Items for Systematic Reviews and Meta-analyses (PRISMA) statement. MEDLINE (via PubMed), Embase, The Cochrane Library, and CINAHL were searched for systematic reviews (SR) published in the period from 2014 to 2019 in the English language. Primary articles cited in SR that fulfil inclusion and exclusion criteria were extracted and examined using narrative synthesis. Interventions were classified based on five chosen elements of the Chronic Care Model (CCM) framework (self-management support, decision support, community resources and policies, delivery system, and clinical information system). Out of 155 SRs retrieved, 7 were considered for the extraction of 166 primary articles. The prevailing setting was the patient’s home. Only 46 studies specified the severity of HF by reporting the level of left ventricular ejection fraction (LVEF) impairment in a heterogeneous manner. However, most studies targeted the populations with LVEF ≤ 45% and LVEF < 40%. Self-management and delivery systems were the most evaluated CCM elements. Interventions related to community resources and policy and advising/reminding systems for providers were rarely evaluated. No studies addressed the implementation of a disease registry. A multidisciplinary team was available with similarly low frequency in each setting. Although HF care should be a multi-component model, most studies did not analyse the role of some important components, such as the decision support tools to disseminate guidelines and program planning that includes measurable targets.

## 1. Introduction

Chronic heart failure (CHF) is a major public health problem because of its high prevalence and complexity. Although the prognosis of CHF has improved, it remains a severe condition with a high frequency of acute decompensations requiring frequent hospitalisations and continuous care imposing complex health needs upon patients.

The prevalence of HF increases with age, ranging from about 1% in those younger than 55 years to more than 10% in those older than 70 years [1,2,3]. However, the true prevalence of heart failure is likely higher since epidemiological studies include only diagnosed cases [4]. The incidence of heart failure in Europe and the USA ranges widely from 1 to 9 cases per 1000 person-years. Among studies limited to older adults, the average incidence reaches 16 cases per 1000 person-years [5,6]. According to recent epidemiological studies conducted in high-income countries, the age-adjusted incidence of HF is decreasing, partly as a result of better management of hypertension and other conditions causing HF. However, with the ageing of the population and increase in hypertension diagnosis, the number of newly diagnosed HF cases increased as well as the number of prevalent cases leading to the increased number of re-hospitalisations, deaths, and overall burden of the disease, imposing an urgent need for reorganisation of current HF management models of care and reprioritisation of resources [7,8].

Reallocation of CHF diagnostic and care to the primary care and community was advocated to improve the care of patients with this chronic disease and multiple comorbidities and to make it more patient-oriented. Programs involving multi-component interventions and multidisciplinary teams represent a recommended strategy to improve outcomes in patients with CHF as they effectively reduce HF hospitalisations, mortality, and all-cause hospitalisations [5]. Moreover, they improve adherence to guidelines and facilitate the approach to complex health and social problems that affect patients and caregivers. There is a vast body of evidence showing the effectiveness of multidisciplinary HF care implemented in various settings and using a range of delivery models, including home-based, clinic-based, and telemonitoring approaches, depending on the patient’s needs, health system organisation, and available resources [9]. Among these strategies, the Chronic Care Model has been defined by the US Health Resources and Service Administration as “a model with key elements of a health care system that encourage high-quality chronic disease care: the community, the health system, self-management support, delivery system design, decision support, and clinical information systems” [10].

Chronic Care Models (CCM) adopting multidisciplinary healthcare programs and diagnostic-therapeutic paths (PDTA) have been shown to be effective in improving health outcomes in different chronic diseases, at least in some studies [11,12,13,14]. However, the evidence about the effectiveness of a CCM approach to HF care is inconclusive [15,16,17,18,19,20,21]. With the exception of self-care interventions [22], it is unknown which elements or combination of CCM elements could improve healthcare practice and health outcomes since there is substantial heterogeneity in the interventions implemented in primary care to improve CHF care delivery [23].

The study aimed to summarise and characterise the interventions used to improve disease management models and clinical pathways in the management of the chronic and acute phases of HF patients and to describe prevalent settings of care, the severity of targeted patients, and the professionals involved.

## 2. Materials and Methods

This scoping review tried to answer the following research question “What are the common features that distinguish and/or unite the different disease management interventions and clinical pathways to manage chronic and acute phases of adult patients with heart failure at different levels of LVEF?” The protocol is available on request from the corresponding author. We have used the PRISMA Extension for Scoping Reviews checklist [24] in the reporting of this review (Appendix A).

### 2.1. Eligibility Criteria

We examined primary articles of the systematic reviews already present in the literature evaluating the effect of disease management interventions and clinical pathways for patients with heart failure in both phases of the disease, acute and chronic. This choice focused on interventions that were submitted to an evaluation and that were considered similar enough to other interventions to be grouped in a systematic review. The PICO(S) framework (Population, Intervention, Comparator, Outcomes, Study design) was used to frame the search strategy (Table 1).

### 2.2. Inclusion and Exclusion Criteria

Due to the broad scope of the review and the substantial number of studies anticipated, only systematic reviews describing multicomponent interventions were considered in the first phase. All primary articles cited in the identified systematic reviews were included in the review.

Studies were relevant for this systematic review if they considered the adult population with HF at any stage of the disease, while patients with cardiac disorders other than HF, with less than 18 years of age, or with congenital HF were excluded. The intervention was any disease management intervention or a clinical pathway used to manage the chronic and acute phases of HF. Studies that did not consider multicomponent interventions were excluded. The comparison group received standard care as defined by the primary studies.

### 2.3. Information Sources, Search Strategy and Selection Process

A search strategy (Appendix A) was developed, including author keywords and database subject headings (MeSH) for three main concepts: heart failure, disease management interventions, and clinical pathways to manage the chronic and acute phases of HF patients.

The search strategy adapted to each database queried was used to search for SRs in the following databases: MEDLINE (via PubMed), Embase, The Cochrane Library, and CINAHL published in the last 5 years in English. The selection by title and abstract of articles to be included for full-text evaluation was carried out by two reviewers (CP, PGR). The selection of full-text articles was carried out by a single reviewer (CP), with a cross-check by another reviewer (OD) on 20% of the selected full-text articles. All inconsistent results were discussed by the reviewers and supervisor (PGR). The study selection process is described in the PRISMA flow chart (Figure 1).

### 2.4. Data Charting Process and Data Items

Data extraction from the full-text articles included in the selected SRs [15,16,17,18,19,20,21] by two reviewers (CP and LM) using a data extraction form. It included the year of publication, the country of publication, the name of the first author, the name of the article, the objective of the study, the study design, the characteristics of the population included (inclusion criteria and sample size), the duration of the study and the follow-up, the description of the intervention, the care settings and the actors involved in the intervention whether health, social or community resources. A shorter version of the extraction form was developed for the description purposes, including study author, year of publication, number of patients overall and in each group, the population included in terms of % of LVEF impairment, aim, intervention, and control description and follow up period, is presented in Appendix A.

### 2.5. Classification of Interventions and Actions

The classification was made by two reviewers (OD and CP) and consequently approved by the supervisor (PGR). The identification of conceptual areas of intervention and their components was based on the Chronic Care Model framework [25] (Table 2). Based on the available literature [26,27,28], the intervention components were classified as relevant for one of the CCM elements: self-management support, decision support, community resources and policies, delivery system, and clinical information system. We did not consider the health system as a separate element, since most of the interventions were carried out in the health system and classifying their components as targeting the health system or not would be arbitrary. A detailed description with examples of intervention components within each CCM element is provided in Table 2. Interventions retrieved were classified by level of LVEF impairment, setting (inpatient, outpatient, primary care, and home), and study size (<100, 100–1000, >1000). Due to heterogeneity in LVEF classification, disease severity was classified for convenience in the following categories based on levels of LVEF impairment:standard or common classification according to the European Society of Cardiology (ESC) guidelines [29]:
-≥50% (normal LVEF or HF with preserved EF (HFpEF))-40–49% (HF with mid-range ejection fraction (HFmrEF)),-<40% (HF with reduced EF (HFrEF)),
other classification containing LVEF cut-offs that overlap with ESC criteria, andnot specified, in case of missing information on LVEF classification.

### 2.6. Synthesis of Results

To better represent the heterogeneity that emerged in the conceptual frameworks of the intervention and its components, the results were aggregated following the Chronic Care Model framework [25]. This model was used for the classification given that at the clinical practice level, five areas or elements of the Chronic Care Model are considered to influence the ability to provide effective chronic disease care: self-management support, delivery system design, decision support, community resources and policies and clinical information systems. Description with definitions and examples of the CCM elements and intervention components are presented in Table 2.

## 3. Results

### 3.1. Selection of Sources of Evidence

Out of 155 unique SRs retrieved, 41 SRs were considered relevant in the screening phase, of which seven SRs [15,16,17,18,19,20,21] were considered for the extraction of primary articles. Overall, 166 unique primary studies were included in this review (Figure 1) [30,31,32,33,34,35,36,37,38,39,40,41,42,43,44,45,46,47,48,49,50,51,52,53,54,55,56,57,58,59,60,61,62,63,64,65,66,67,68,69,70,71,72,73,74,75,76,77,78,79,80,81,82,83,84,85,86,87,88,89,90,91,92,93,94,95,96,97,98,99,100,101,102,103,104,105,106,107,108,109,110,111,112,113,114,115,116,117,118,119,120,121,122,123,124,125,126,127,128,129,130,131,132,133,134,135,136,137,138,139,140,141,142,143,144,145,146,147,148,149,150,151,152,153,154,155,156,157,158,159,160,161,162,163,164,165,166,167,168,169,170,171,172,173,174,175,176,177,178,179,180,181,182,183,184,185,186,187,188,189,190,191,192,193,194,195].

### 3.2. Characteristics of Included Studies

Characteristics of the included studies are reported in Appendix A. Of the 166 studies included, 161 had different quantitative study designs (139 RCT, 1 non-randomised trial, 9 cohort studies, 12 other study designs), and five were qualitative studies. Most of the evaluated interventions were implemented in North America (USA: 73, Canada: 6), followed by EU countries (63, of which 11 were from the UK), Australia (7), Asia (11), and South America (6). The sample size ranged from 10 to 3031, with 13 studies including more than 1000 patients. Studies with small samples were usually conducted in academic settings, while larger samples were used in studies that usually evaluate the efficacy or feasibility of the intervention. The home was most frequently the setting of the intervention (118), followed by equally represented inpatient and outpatient clinics (56 and 53 studies, respectively), while only 13 studies evaluated interventions in the primary care setting.

Out of 166 studies included, only 46 specified the severity of HF by reporting the level of LVEF impairment. The majority of studies included HF patients with HFrEF (LVEF < 40%) (n = 19), while only four [41,49,50,122] and one study [154] considered patients with HFmrEF (LVEF 40–49%) and HFpEF (LVEF ≥ 50%), respectively. Twenty-two studies examined specific target populations with LVEF range that overlap with the ESC classification, such as LVEF ≤ 55% (3 studies) [107,177,178], LVEF ≥ 45% (4 studies) [86,128,141,146], and LVEF ≤ 45% (21 studies) [36,41,44,49,50,73,80,91,93,115,122,126,136,141,143,145,146,155,163,177,180].

### 3.3. Description of the Interventions

Overall, self-management (152 studies) and delivery system interventions (132 studies) were the mostly evaluated CCM elements, followed by decision support (69 studies) and clinical information system (53 studies) (Table 3). Interventions related to community resources and policy were subject to evaluation only in 7 out of 166 included studies. Face-to-face education, self-monitoring and medical management tools, and mHealth education were the most analysed aspects of self-management support, while eHealth education and physical activity were the least represented ones. Telemedicine/remote monitoring and advanced practitioner nurse involvement were the two most evaluated components of the delivery system. All three aspects of the decision support (integrated CHF protocols into routine practice, provider education, and linkage between primary and speciality care) were equally represented. Monitoring indicators and feedback to providers and sharing information between providers were the two predominant components of the clinical information system element. Only eight of 53 studies assessed the effectiveness of the advising/reminding system for providers, while no studies addressed the implementation of a disease registry. Few studies that evaluated community resources and policy focused mostly on social support while linking patients to outside resources, logistic support, third-sector involvement, and community-based self-management programs were even less represented.

As summarised in the Venn diagram (Figure 2), most of the interventions covered more than one CCM element, but only one study [40] combined components from all CCM elements under study (self-management support, delivery system, decision support, community resource and policy, and clinical information system). Self-management support interventions were frequently analysed in the presence of a delivery system (telemedicine/remote monitoring) (120 studies), decision support (66 studies), or clinical information system interventions (55 studies). Less frequently, decision-support interventions were combined with delivery system interventions (58 studies).

### 3.4. Type of Intervention Components by Level of LVEF Impairment, Setting, and Size

When considering studies that specified HF severity, populations with LVEF ≤ 45% and LVEF < 40% were analysed the most (Table 3). Consequently, all CCM elements and interventions were mostly analysed in these groups of patients.

Self-management support, decision support and delivery system were mostly implemented in the home setting and to a lesser extent in inpatient and outpatient setting, while clinical information system was predominantly related to home and outpatient care due to the collection and processing of clinical information data within the remote monitoring (Table 4). Interestingly, provider education was frequently offered to health workers conducting interventions in the home setting, while workers in other health settings were less subject to educational interventions. Only half (23 out of 56) studies conducted in inpatient settings offered a discharge/care planning intervention.

All CCM elements were predominantly analysed in studies with medium sample sizes (100–1000 patients) and in home settings (Table 5). Studies with more than 1000 patients were conducted mostly in home settings and analysed self-management interventions, delivery systems, and clinical information systems.

### 3.5. Team Organisation Structure by Setting and LVEF Impairment

Interventions with the multidisciplinary team were available with similarly low frequency in each setting, except the primary setting, where only 4 out of 59 studies with multidisciplinary intervention were conducted (Table 4).

Nurses and cardiologists were the most involved professionals in all settings (Table 6). Around half of the included studies had APC nurse involved, while nurse case manager was involved in 53 studies. They were involved mostly in studies conducted in the home and outpatient settings. In all cases, they were responsible for coordinating and managing care, supporting patient self-care, and ensuring that planned follow-ups were carried out. Other professionals, such as pharmacists, nutritionists, physiotherapists, social workers, physicians, geriatricians, psychologists, and psychiatrists, were considered in multidisciplinary care but not as a part of the dedicated team.

Studies that included patients with HF with reduced EF (<40% LVEF) involved mostly cardiologists, physicians and APC nurses, while nurse case managers and other health specialists were less often considered.

### 3.6. Qualitative Studies

All five qualitative studies included [99,114,125,167,185] were conducted with a phenomenological approach. The common objective of these studies was to investigate the most commonly perceived barriers to self-care management. Lack of awareness, depression, weight problems, difficulty in exercising, fatigue, poor communication with the doctor, and poor family support, are the most frequently detected obstacles to self-management. The self-care regimen CHF was perceived by both patients and physicians as work, but patient-physician dyads show divergent interpretations of such labour. Physicians perceived patients as not participating enough in self-care despite they considered instructions being “easy”. Patients perceived themselves as being able to understand what to do but needing help on how to perform self-care.

## 4. Discussion

We carried out a scoping review of 166 primary articles cited by the 7 SRs to understand better which interventions proposed and evaluated so far have been used to improve disease management models and clinical pathways of the chronic and acute phases of HF. The results were categorised and interpreted following five CCM elements (self-management support, decision support, community resources and policies, delivery system, and clinical information system). Overall, self-management interventions (face-to-face education, self-monitoring and medical management tools and m-health education) and delivery system interventions (telemedicine/remote monitoring and advanced practitioner nurse involvement) were the mostly evaluated CCM elements, while interventions related to community resource and policy were rarely evaluated, as well as advising/reminding system for providers. No studies addressed the implementation of a disease registry. Only one study evaluated all five CCM elements considered in this study [40].

The studies were carried out in different healthcare contexts; nevertheless, some common concepts emerged. The prevailing management setting investigated was the patient’s home, given that self-management interventions were the most evaluated CCM elements. Actions to improve or support self-management, such as patient and/or caregiver education, were frequently analysed in the presence of changes in the delivery system, in particular the introduction of telemonitoring, and less frequently in the implementation of clinical information system interventions (monitoring indicators and feedback to provider. Such a combination of interventions was predominantly conducted in the home setting and delivered by APN nurses. Self-care interventions are mainly used in the population with LVEF ≤45%, as well as for interventions referred to the other intervention areas provided by the CCM. The severity of HF was classified in a heterogeneous manner in the retrieved studies, and only in some cases %LEVF was specified. Greater clarity and harmonisation of HF severity classification are needed to understand which intervention to prioritise according to the severity of HF [9].

Self-care support can be offered to individual patients, the patient-caregiver dyad, or groups of patients through mHealth and eHealth educational interventions on self-monitoring and medical management or through face-to-face didactic sessions by educators using printed or written materials. Educational interventions using eHealth or web approach are less represented, although they could improve healthcare accessibility and overcome geographic inequalities as well as organisational challenges for families and caregivers. Furthermore, the impact on health inequalities of interventions based on mHealth needs to be carefully assessed. In fact, despite the fact that mHealth gives great opportunities given the high penetrance of smartphones in all socio-economic strata and educational levels of the population, it also may introduce barriers to access in those HF patients, usually the oldest and most socially fragile ones, who have low digital literacy.

Discharge planning and follow-up monitoring remain fundamental steps to assure a continuum of care between hospital and primary care management of patients with heart failure. Emphasis is placed on patient/caregiver education as a fundamental intervention of the care pathway, and post-discharge monitoring frequently includes checks on acquired educational notions and reinforcement interventions aimed at increasing self-care and self-monitoring skills. Yet only half of the included studies conducted in inpatient settings offered a discharge/care planning intervention, and discharge planning was rarely analysed together with patient or caregiver education.

When the structure of the care team (physician, nurse, etc.) was studied, nurses and cardiologists were the most frequently involved professionals in all settings, followed by nutritionists and pharmacists. A multidisciplinary team was considered in only one-third of studies that evaluated delivery systems. Multidisciplinarity was the least evaluated in the primary care setting; this may be because the multidisciplinary team in a hospital setting is already a well-established standard, while in outpatient and home care, it is not. Interestingly, health provider education was mostly offered to health workers conducting interventions in the home setting, while workers in other health settings were less frequently targeted by educational interventions. This suggests that the transition of CHF care to primary care in terms of setting and the professionals involved has not been fully developed despite suggestions and efforts [29].

In most cases, multidisciplinary consultation was accompanied and facilitated by the presence of an advanced practice nurse and less frequently by the nurse case manager, although their function was mentioned with different terms (care coordination, nurse management, nurse-led care) but with similar tasks. The advanced practice nurse in the literature does not have a universally accepted definition, as well as the required skills and the level of advanced training required are often not described, despite having an important role in supporting patient self-care and ensuring the planning and conduct of patient follow-up as required by care plans.

Clinical information systems and decision support tools to facilitate the application of the guidelines on which the model is based by healthcare professionals are less represented in the literature. Telemedicine and quality improvement measures and monitoring need specific information systems. In the retrieved studies, the development of information systems was reported mainly when telemedicine interventions were included, while interventions to improve connections between health providers and between health providers and patients are less represented. Feedback to professionals showing their performance levels against chronic disease indicators and implementation of disease registries were not evaluated at all. The role of clinical information systems has been underestimated or not emphasised in the studies evaluating interventions to improve the new management of chronic diseases in primary care despite its well-recognised role in planning appropriate care for patients with different comorbidities [196].

In the retrieved studies, the involvement of community resources was scarcely considered. In the few studies involving resources outside the health system and the patient’s family, these are mostly considered for supporting self-help groups involving peer leaders and student volunteers. We did not find studies evaluating a deeper involvement of public services not related to the health system, nor the involvement of informal social networks to reduce logistical barriers for patients and to sustain caregivers. Despite there is evidence that community involvement can help patients and caregivers be more compliant with certain cues (facilitating travel, helping time balance for caregivers) and can facilitate healthy lifestyle choices [40,45,164], we must note that there are very few experiences reported in the scientific literature for chronic care of HF. Our results should be read in light of some limits. We only tried to describe the main components of the interventions employed to improve disease management models and clinical pathways in the care of the chronic and acute phases of HF patients. Therefore, we decided not to evaluate the quality of the included studies. Furthermore, we could not evaluate the proposed models for their feasibility nor if they have been actually implemented or were just experimented with in academic settings. Moreover, we have described interventions with respect to the CCM, which is considered a high-quality approach to traditional HF management; however, focusing on the complex intervention, we probably missed some of the most innovative parts of HF management, in particular, precision cardiology approaches which use clinical and genetic characteristics of the individual to define personalised and precise disease management [197]. These limits are relevant to the scope of our review, and they should be carefully considered when using our results to construct a new model or to start a systematic review to assess the efficacy of specific components or types of care models. Research through other database analyses and grey literature may have yielded other relevant articles. In addition, because the review was limited to papers published in the English language, it is possible that other potentially relevant articles and reviews were omitted. Nevertheless, including more than 160 studies guaranteed a saturation of the different components of the interventions proposed for the management of HF patients, which was the main goal of our search strategy.

This study has some important implications for future research and clinical practice. The combination of telemedicine and clinical decision support systems is rarely evaluated together despite being essential in enabling physicians to promptly adapt medication doses and, therefore, reducing the number of hospital visits needed. In addition to this, tools to support the adoption of evidence-based guidelines should be evaluated and implemented in practice. The development of eHealth and telemedicine is a very promising area that would merit more in-depth research and development efforts in the future, particularly because of its potential to reduce the burden of self-management on patients and caregivers. Finally, program planning that includes measurable targets for better HF care, which is recommended by the CCM, but scarcely reported in the literature, should become part of health system priorities to support the new management of chronic diseases. If this does not happen, innovations in care processes are unlikely to be introduced and even more unlikely that the quality of care will be rewarded.

## 5. Conclusions

There is great heterogeneity in the classification of heart failure severity used to target patients. This heterogeneity makes it difficult to understand which HF patients could benefit from interventions and their components and if some interventions could be implemented to a wide range of severity, and which are more focused.

Although all CCM components of interest (patient self-care support, delivery system, decision support, community resource and policy, clinical information system) are represented in the literature, only one study integrated all the conceptual domains related to the CCM interventions for the care of patients with heart failure. This probably reflects the difficulties in evaluating complex interventions but may also reflect the difficulties in implementing interventions simultaneously acting on different aspects of the health system, the community, the patient, and the professionals.

## Figures and Tables

**Figure 1 healthcare-11-01227-f001:**
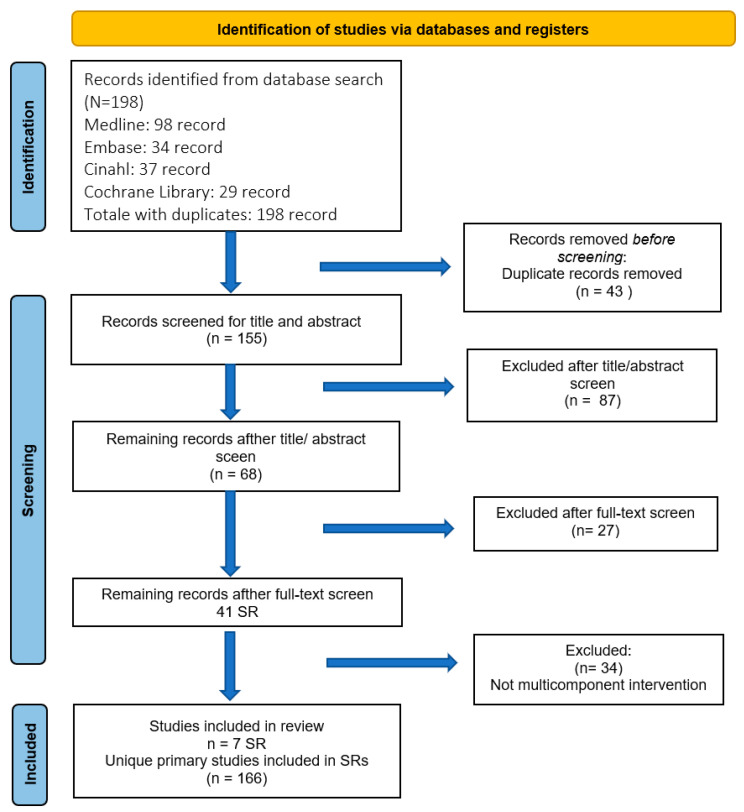
Flowchart of study selection.

**Figure 2 healthcare-11-01227-f002:**
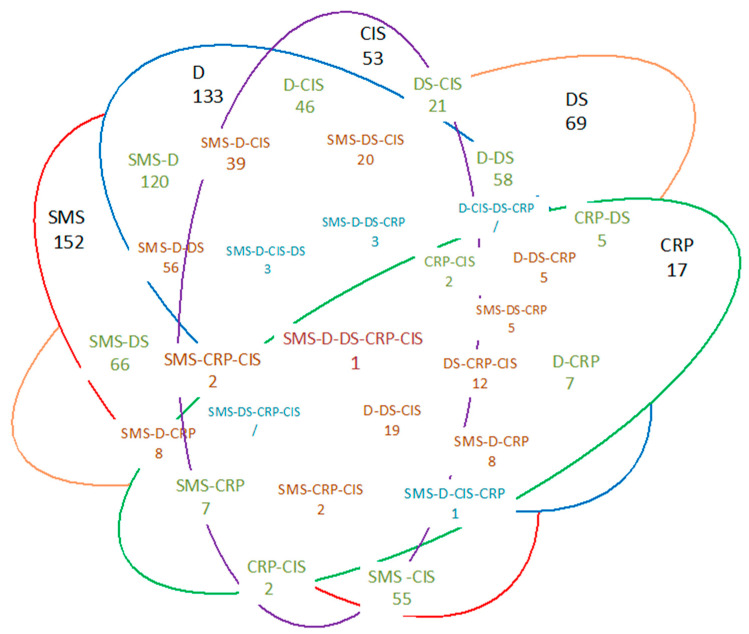
Venn diagram of intervention components. D, delivery system; DS, decision support; CIS, clinical information system; SMS, self-management support; CRP, community resource, and policy.

**Table 1 healthcare-11-01227-t001:** PICO question.

Description	Scope
Population	Adult patients with heart failure at any stage of the disease
Intervention	Multicomponent disease management interventions and clinical pathways to manage the chronic and acute phases of HF patients
Comparator	Standard care (routine or standard care, as defined by the primary studies)
Outcomes	Studies investigating any outcome of efficacy, effectiveness, and costs will be considered
Study design	All study designs were included, given the broad scope of the review. No limits were given on the duration of the intervention or the length of follow-up.

**Table 2 healthcare-11-01227-t002:** Classification of conceptual areas of intervention disease management and its components based on the Chronic Care Model framework [25].

CCM Element	Description of the CCM Element	Intervention Components	Description and Example
Self-management support	Emphasis on the importance of the central role the patients have in managing their own care.	Educational interventions	Educational interventions on self-monitoring, medical management, decision making, or adoption and maintenance of health-promoting behaviours, divided into: - mHealth-based interventions (delivery of health messages, interventions, and verification of notions provided through education via mobile phones, tablets, and other wireless technologies), - eHealth (web-based computer-tailored interventions) and - face-to-face teaching sessions conducted by educators using written or printed materials
		Motivational counselling and/or behavioural therapy/support	Telephonic or face-to-face motivational counselling sessions focused on self-monitoring and medical management, decision-making, or adoption and maintenance of health-promoting behaviours.
		Family and caregiver education/support	Any kind of educational, motivational, behavioural intervention oriented towards a family member or caregiver.
		Physical activity	Provision of individual or group physical activity lessons, instructions, or programs.
		Self-monitoring and medical management tools	Distributed logs, notebooks, calendars, and dosette boxes or provided technological aids (electronic reminders, phone cues) for self-monitoring, for example, salt intake or weight control.
		Telephone advice lines	Working hours or out-of-hours answerphone system providing advice/support service about self-management.
Decision support	Integration of evidence-based guidelines into daily clinical practice	Integrated CHF protocols into routine practice	Implementation of protocols or guidelines into daily clinical practice.
		Provider education	Any kind of education or case discussions with care providers, usually nurses.
		Linkages between primary and speciality care	Organisation or coordination of patient care activities and sharing of clinical information between different professionals involved in primary care and speciality health services.
Community resources and policies	Developing partnerships with community organisations that support and meet patients’ need	Linking patients to outside resources	Referring a patient to a local community health program, church-based support groups, and clinic-based support groups.
		Logistic support	Providing transport to patients from home to the outpatient clinic or community intervention site.
		Third sector involvement	Activities with community-based organisations, volunteer groups, self-help groups, centres for the elderly, etc.
		Community-based self-management programs	Group intervention attended in the community aims to improve disease control and promote self-efficacy.
		Social support	Social support provided by community-based organisations or involvement in social structures within the community.
Delivery system	Focus on teamwork and an expanded scope of practice for a team member to support chronic care	Patient care planning/discharge planning	Development of an individualised discharge plan or adaptation of recommendations and prescriptions for a patient prior to their discharge from the hospital.
		Telemedicine/remote monitoring	Use of telecommunication equipment to remind patients or detect early signs and symptoms of heart failure.
		Multidisciplinary team	Involvement of three or more providers from different healthcare specialities in patient care.
		Advanced practice nurse involvement	Advanced practice nurses are involved in the provision of care services.
		Nurse-led/Nurse case manager	The activities of management, assessment, planning, and coordination of patient care are carried out under the responsibility of the nurses.
Clinical information system	Developing information systems based on patient populations to provide relevant client data	Disease registry	Computer or web-based applications or systems used to capture, manage, and provide information about the specific condition to support organised care management of patients.
		Monitoring indicators and feedback to the provider	Collecting and sharing biometric data and patient-reported insights with care teams who evaluate trends and intervene, if necessary.
		Advising/reminders systems for providers	E-mails or messages sent to nurses that contain reminders, instructions, and/or guidelines.
		System for sharing information between providers	Web-based medical records accessible to all health professionals involved in the care of the patient.

**Table 3 healthcare-11-01227-t003:** The number of studies by type of setting, type of intervention, and HF severity (level of LVEF impairment).

			Severity (LVEF)							
			Not Specified n = 120	ESC Classification			Other Classification			Overall n = 166
				≥50% N = 1	40–49% N = 4	<40% N = 19	≥45% N = 4	≤45% N = 15	≤55% N = 3	
Setting		Inpatient	44	1	1	6	/	4	/	56
		Outpatient	34	1	/	10	/	7	1	53
		Primary care	9		/	1	/	3	/	13
		Home	87	1	3	13	4	7	3	118
Intervention	Self-management support	m-Health education	57	/	2	8	3	5	/	74
		e-Health education	17	/	/	5	/	1	/	23
		Face-to-face didactic session	84	1	3	12	4	13	3	120
		Motivation counselling and/or behavioural therapy/support	35	1	/	6	2	3	/	47
		Family and caregiver education/support	36	/	2	4	3	8	3	56
		Physical activity	8	/	/	1	2	2	1	15
		Self-monitoring and medical management tools	62	1	1	6	3	6	/	79
		Telephone advice lines	41	1	/	5	/	3	2	52
		Overall	109	1	4	16	4	15	3	152
	Decision support	Integrated CHF protocols into routine practice	28	/	1	2	/	2	/	33
		Provider education	18	/	/	2	2	3	/	25
		Linkage between primary and speciality care	19	/	/	5	/	3	2	29
		Overall	50	0	1	9	2	5	2	69
	Community resource and policy	Linking patients to an outside resource	4	/	/	/	/	1	/	5
		Logistic support		/	/	1	/	1	/	2
		Third sector involvement	1	/	/	/	/	1	/	2
		Community-based self-management programs	2	/	/	/	/	/	/	2
		Social support	8	/	/	1	/	3	/	12
		Overall	5	0	0	1	0	1	0	7
	Delivery system	Patient care planning/discharge planning	25	/	/	2	/	2	/	29
		Telemedicine/remote monitoring	56	/	2	12	/	2	3	75
		Multidisciplinary team	23	/	/	5	/	3	1	32
		Advanced practitioner nurse involvement	37	1	2	7	3	7	1	58
		Nurse-led/nurse case manager	25	/	/	4	/	4	2	35
		Overall	97	1	3	14	3	11	3	132
	Clinical information system	Disease registry	/	/	/	/	/	/	/	/
		Monitoring indicators and feedback to the provider	21	/	1	12	/	1	1	36
		Advising/reminding system for providers	7	/	/	1	/	/	/	8
		Sharing information between providers	12	/	2	3	/	2	1	20
		Overall	35	0	2	12	0	2	2	53

**Table 4 healthcare-11-01227-t004:** Intervention type by setting.

		Inpatient n = 56	Outpatient n = 53	Primary Care n = 13	Home n = 118	Overall n = 166
Self-management support	m-Health education	23	17	6	58	104
	e-Health education	6	8	1	19	34
	Face-to-face didactic session	49	43	10	84	186
	Motivation counselling and/or behavioural therapy/support	21	14	4	35	74
	Family and caregiver education/support	23	17	5	45	90
	Physical activity	3	4	2	8	17
	Self-monitoring and medical management tools	33	26	8	58	125
	Telephone advice lines	28	16	3	41	88
	Overall	53	49	12	108	222
Decision support	Integrated CHF protocols into routine practice	11	13	4	22	50
	Provider education	7	5	2	14	28
	Linkage between primary and speciality care	9	12	2	19	42
	Overall	22	23	6	45	76
Community resource and policy	Linking patients to outside resources	1	2	1	2	6
	Logistic support	/	1	1	1	3
	Third sector involvement	1	/	1	1	3
	Community-based self-management programs	1	1	/	1	3
	Social support	/	/	/	/	/
	Overall	2	3	1	4	10
Delivery system	Patient care planning/discharge planning	23	6	1	26	56
	Telemedicine/remote monitoring	26	20	5	64	115
	Multidisciplinary team	17	14	4	19	59
	Advanced practitioner nurse involvement	17	22	3	42	83
	Nurse-led/nurse case manager	12	14	3	29	58
	Overall	49	40	8	101	198
Clinical information system	Disease registry	/	/	/	/	/
	Monitoring indicators and feedback to the provider	8	13	/	30	51
	Advising/reminding system for providers	1	3	/	5	9
	Sharing information between providers	4	5	2	15	26
	Overall	11	17	4	40	72

**Table 5 healthcare-11-01227-t005:** Intervention type by study size.

				Study Size			
				<100 n = 35	100–1000 n = 118	>1000 n = 13	Overall n = 166
Setting		Inpatient		8	36	6	50
		Outpatient		11	36	3	50
		Primary care		2	10		12
		Home		26	77	11	114
Intervention	Self-management support	m-Health education	19		47	6	72
		e-Health education	5		15	2	22
		Face-to-face didactic session	22		83	7	112
		Motivation counselling and/or behavioural therapy/support	11		28	6	45
		Family and caregiver education/support	9		39	5	53
		Physical activity	4		8	/	12
		Self-monitoring and medical management tools	15		51	6	72
		Telephone advice lines	6		37	5	48
		Overall	32		102	10	142
	Decision support	Integrated CHF protocols into routine practice	3		23	3	29
		Provider education	5		15	3	23
		Linkage between primary and speciality care	1		23	4	28
		Overall	9		47	7	63
	Community resource and policy	Linking patients to an outside resource	/		4	/	/
		Logistic support	/		12	/	/
		Third sector involvement	/		1	/	/
		Community-based self-management programs	/		/	/	/
		Social support	/		/	/	/
		Overall	/		5	/	5
	Delivery system	Patient care planning/discharge planning	3		19	4	26
		Telemedicine/remote monitoring	13		54	6	73
		Multidisciplinary team	6		21	3	30
		Advanced practitioner nurse involvement	13		37	7	57
		Nurse-led/nurse case manager	20		57	8	85
		Overall	28		85	12	125
	Clinical information system	Disease registry	/		/	/	/
		Monitoring indicators and feedback to the provider	8		24	3	35
		Advising/reminding system for providers	3		2	2	7
		Sharing information between providers	3		15	2	20
		Overall	11		35	15	61

**Table 6 healthcare-11-01227-t006:** Professionals involved by type of setting and HF severity.

	Setting					HF Severity (LVEF)						
	Inpatient n = 56	Outpatient n = 53	Primary Care n = 13	Home n = 118	Overall	≥50% N = 1	40–49% N = 4	<40% N = 19	≥45% N = 4	≤45% N = 15	≤55% N = 3	Overall
Advance practice nurse	19	22	3	44	88	1	2	7	3	7	1	21
Nurse-led	10	14	3	26	53	/		4		4	2	10
Nurse	29	19	6	50	104	/	1	6	1	7	1	16
Cardiologist	19	22	7	32	80	/	2	10	/	2	1	15
Geriatrician	2	1	/	1	4	/	/	/	/	/	/	/
Pharmacist	7	4	3	13	27	/	/	2	/	2	1	5
Physician	/	1	/	1	2	/	/	7	/	2	1	10
Psychiatrist	2	/	1	1	4	/	/	1	/	3	/	4
Psychologist	2	2	2	1	7	/	/	/	/	/	/	/
Physiotherapist	1	/	/	/	1	/	/	/	/	1	/	1
Dietist/nutritionist	12	6	2	12	32	/	/	/	/	2	/	2
Social worker	7	5	1	9	22	/	/	1	/	/	/	1
Occupational therapist	/	/	1	/	1	/	/	/	/	/	1	1
Students pursuing premedical track	1	/	/	1	2	/	/	/	/	/	/	/
Patients	/	/	/	1	1	/	/	/	1	/	/	1

## Data Availability

This is a scoping review of peer-reviewed scientific literature. Data used came from scientific manuscripts, which can be accessed online. All relevant information is included in the manuscript.

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
