# Peer review of "Elements Characterising Multicomponent Interventions Used to Improve Disease Management Models and Clinical Pathways in Acute and Chronic Heart Failure: A Scoping Review"

_healthcare, 2023, doi:10.3390/healthcare11091227_

Round 1
Reviewer 1 Report
The search strategy is clearly outlined, including the use of author keywords and MeSH terms to identify relevant articles from selected databases. The process for selecting articles for inclusion is also described in detail, which involves screening by title and abstract followed by a full-text evaluation by two reviewers.
The data charting process is also clearly outlined, including the data items that were extracted from the included studies.
-
The abstract could be more concise. Some of the sentences are quite long and could be broken up to improve readability.
-
The study does not clearly state the main findings of the study. It would be helpful to include a sentence or two summarizing the most important results, such as which interventions were found to be most effective.
-
The study could benefit from more detailed information about the inclusion and exclusion criteria used to select primary articles for examination. This would provide more context for understanding the study's methods.
-
The study could also benefit from a more explicit discussion of the implications of the findings for clinical practice or future research.
Author Response
REVIEWER 1:
The search strategy is clearly outlined, including the use of author keywords and MeSH terms to identify relevant articles from selected databases. The process for selecting articles for inclusion is also described in detail, which involves screening by title and abstract followed by a full-text evaluation by two reviewers. The data charting process is also clearly outlined, including the data items that were extracted from the included studies.
RE: We thank the reviewer for the comments.
The abstract could be more concise. Some of the sentences are quite long and could be broken up to improve readability.
RE: We have broken the long sentences to improve readability.
The study does not clearly state the main findings of the study. It would be helpful to include a sentence or two summarizing the most important results, such as which interventions were found to be most effective.
RE: Given the scoping design of this review, we were not able to assess the effectiveness of interventions, only to describe which of them have been evaluated so far and with respect to which setting and professionals included. We have tried to summarize more explicitly the main findings in the first paragraph of the Discussion, although it is very challenging given their extent and complexity. The main results regarding the frequently evaluated interventions with respect to setting and the patterns of their combinations within a single management program are summarized in the second paragraph of the Discussion.
The study could benefit from more detailed information about the inclusion and exclusion criteria used to select primary articles for examination. This would provide more context for understanding the study's methods.
RE: We have added a paragraph that describes inclusion and exclusion criteria.
The study could also benefit from a more explicit discussion of the implications of the findings for clinical practice or future research.
RE: We have added a paragraph with implications for future research and clinical practice.

Reviewer 2 Report
This is a nicely written review about the current state-of-the-art for disease management in reference to heart failure.
Authors utilized the PRISMA guidelines to search the available literature and summarize the approaches in a systematic manner. For the most part, I am satisfied with the manner in which the review is written. I just have a couple of points below that might further improve the usage of this review:
1) There are multiple precision cardiology approaches being utilized (at least in US) where the genetic constitution of an individual can be used for disease management. For example: https://www.broadinstitute.org/precision-cardiology-laboratory
It could be interesting if authors could mention this somewhere, either in the discussion or as additional paragraph.
2) Lines 35-37: To an uninitiated reader, the 5 chose elements of CCM might come across as unfamiliar. Please provide suitable reference here or add a line explaining how well defined this is.
3) Lines 112-116: Defined keywords were employed. How were synonymous words to these keywords dealt with? Was the search completely manual or it had an automated component at some stage?
4) Lines 117 onwards: Were only reviews searched? What about original research articles discussing novel disease management strategies or case studies for HF?
Discussion is well written, except if authors could stress the main points that need to be included in HF disease management (and are missing currently) based on their scoping, that could be a good take home message. For instance, telehealth and clinical decision support systems appear essential, but less common.
Author Response
REVIEWER 2:
This is a nicely written review about the current state-of-the-art for disease management in reference to heart failure. Authors utilized the PRISMA guidelines to search the available literature and summarize the approaches in a systematic manner. For the most part, I am satisfied with the manner in which the review is written. I just have a couple of points below that might further improve the usage of this review:
1) There are multiple precision cardiology approaches being utilized (at least in the US) where the genetic constitution of an individual can be used for disease management. For example: https://www.broadinstitute.org/precision-cardiology-laboratory
It could be interesting if authors could mention this somewhere, either in the discussion or as additional paragraph.
RE: Thank you for this interesting comment. Given that precision cardiology approaches did not emerge from the systematic review, we mentioned this as a limit of our review, i.e. it was not able to identify some innovative care models.
2) Lines 35-37: To an uninitiated reader, the 5 chose elements of CCM might come across as unfamiliar. Please provide suitable reference here or add a line explaining how well defined this is.
RE: We have tried to explain it better.
3) Lines 112-116: Defined keywords were employed. How were synonymous words to these keywords dealt with? Was the search completely manual or it had an automated component at some stage?
RE: We use a specific search strategy (vocabulary and free text) and a specific filter for systematic review because of a consistent amount of literature.
4) Lines 117 onwards: Were only reviews searched? What about original research articles discussing novel disease management strategies or case studies for HF?
RE: Due to the broad scope of the review and a substantial number of studies, only systematic reviews describing multi professional and multicomponent interventions were considered at the first phase. All primary articles cited in the identified systematic reviews were included in the review. Case studies were excluded from the reviews retrieved and therefore from our study as well.
Discussion is well written, except if authors could stress the main points that need to be included in HF disease management (and are missing currently) based on their scoping, that could be a good take home message. For instance, telehealth and clinical decision support systems appear essential, but less common.
RE: We have added a paragraph with implications for future research and clinical practice.
